# Collective memory for American leaders: Measuring recognition for the names and faces of the US presidents

**Adam L. Putnam** [1]*, **Sarah Madison Drake**[1], **Serene Y. Wang**[1], **K. Andrew DeSoto**[2]

1 Department of Psychology, Furman University, Greenville, South Carolina, United States of America,
2 Association for Psychological Science, Washington, DC, United States of America

* adam.putnam@furman.edu

**Data Availability Statement:** Data and Materials for E1 are available at OSF.IO/7EZ4F. Data and Materials for E2 are available at OSF.IO/6TWDH.

## Abstract

Collective memory studies show that Americans remember their presidents in a predictable pattern, which can be described as a serial position curve with an additional spike for Abraham Lincoln. However, all prior studies have tested Americans' collective memory for the presidents by their *names*. How well do Americans know the *faces* of the presidents? In two experiments, we investigated presidential facial recognition and compared facial recognition to name recognition. In Experiment 1, an online sample judged whether each of the official portraits of the US presidents and similar portraits of nonpresidents depicted a US president. The facial recognition rate (around 60%) was lower than the name recognition rate in past research (88%), but the overall pattern still fit a serial position curve. Some nonpresidents, such as Alexander Hamilton, were still falsely identified as presidents at high rates. In Experiment 2, a college sample completed a recognition task composed of both faces and names to directly compare the recognition rates. As predicted, subjects recognized the names of the presidents more frequently than the faces. Some presidents were frequently identified by their names but not by their faces (e.g. John Quincy Adams), while others were the opposite (e.g. Calvin Coolidge). Together, our studies show that Americans' memory for the faces of the presidents is somewhat worse than their memory for the names of the presidents but still follows the same pattern, indicating that collective memories contain more than just verbal information.

## Introduction

When looking at the front page of a newspaper or the homepage of a news website, it is difficult to not find at least one mention of the President of the United States of America. Stepping into the role of president immediately thrusts an individual to an almost unparalleled level of fame around the world. Virtually every American, as well as many others, now recognizes their name and appearance. Such fame, however, does not last forever.

The prestige and the widespread knowledge of the US president make it a prime example of collective memory. Collective memories are memories shared by a group, often ones that

**Funding:** The authors received no specific funding for this work.

shape the group's identity [1]. Collective memory scholars have studied various aspects of collective memory, including the contents of shared memories, how shared memories shape a group's identity, and how these shared memories change over time [2]. Although collective memory has long been studied qualitatively by scholars in various disciplines, empirical research on how institutions, as well as individuals within institutions form collective memories did not emerge until recent years [1]. One line of collective memory research studies patterns of shared knowledge formed by people in large groups, such as Americans' common knowledge about their current and past presidents.

Previous research has shown that Americans remember their presidents in predictable patterns across time and generations. In general, the recall of the presidents fits a serial position curve, where the first few presidents, the most recent presidents, and presidents associated with distinctive historical events are remembered best [3–5]. More specifically, Roediger and DeSoto [5] reported three experiments with samples across 35 years where they examined Americans' recall for the US presidents. They found that Americans had nearly perfect recall for recent presidents, with predictable and systematic forgetting of the older presidents, except for George Washington (the first president) and Abraham Lincoln (associated with the Civil War), who were also recalled nearly perfectly. Similar serial position curves of collective memory for a country's leaders have also been demonstrated in China [6] and Canada [7]. But countries without single leaders may not show the same pattern (e.g., Swiss participants show a recency but no primacy effect [8]).

Americans also show predictable patterns in their ability to *recognize* the names of the presidents [9], although they are better at recognizing presidents (.88 of presidents recognized) than recalling them (.43 recalled in Roediger & DeSoto [5]). Critically, the memory patterns are similar in both types of test: excellent memory for the earliest presidents (a primacy effect), the latest presidents (a recency effect), and Lincoln. Curiously, the recognition task also revealed that some nonpresidents, such as Alexander Hamilton, were falsely recognized as presidents more often than some actual presidents.

These studies, however, have a limitation. Memory for the presidents was always evaluated by having Americans respond to the presidents' *names*, raising the question of whether Americans would show similar patterns of memory for the presidents' *faces*. While we can predict that most Americans would recognize pictures of Donald Trump or Joe Biden, it is unclear whether Americans would also recognize pictures of Jimmy Carter or John F. Kennedy, despite being likely to recognize their names as being those of presidents. This raises the question of whether a president's name or picture will be easier to recognize.

On one hand, a huge amount of research has demonstrated the picture superiority effect: pictures and images are remembered better than words in nearly all cases [10]. The picture superiority effect is prevalent in multiple types of memory tasks [11], including free recall, serial recall and reconstruction [10], associative recognition [12], and recognition [13]. In one striking example of the picture superiority effect, Standing [14] showed that subjects could recognize 88% of pictures two days after viewing each of them for 5 sec. The impressive feature of this study is that subjects studied 1,000 pictures! Therefore, Americans may better recognize the faces than the names of the presidents either because of the increased distinctiveness of pictures or because pictures may be encoded both as an image and as an associated verbal label (dual coding theory; [10]).

On the other hand, Americans may not recognize the faces of the presidents as well as their names. One possibility here is that presidents are typically referred to by their names in historical documents and texts. Repeated exposure to the names of past presidents without reference to their faces may then lead to better recognition of the names compared to the faces.

The goal of the current study was to examine how well Americans recognize the faces of the presidents and how facial recognition compares to name recognition. This question is important because the majority of collective memory research (at least within psychology) is verbal in nature. Additionally, we were interested in whether confidence in presidential recognition decisions would be similar for faces and names. In Experiment 1, subjects completed a facial recognition task consisting of portraits of US presidents, vice presidents, and other lures. In Experiment 2, subjects completed a recognition task where people were presented as portraits or names. We expected that there would be a reasonably strong correlation between the recognition rates for names and faces and that both types of test would show serial position curves. However, we also expected that faces would be remembered less often than names.

## Experiment 1

We preregistered Experiment 1 on the Open Science Framework at OSF.IO/7EZ4F where we reported all conditions, measures, and analysis plans. In Experiment 1, subjects saw 132 portraits and indicated whether each face was of a president and rated their confidence in their decision. We predicted that the pattern of recognition of the faces would be generally similar to that of names in prior research, but less accurate [9]. We expected that subjects would be more likely to recognize presidents than to falsely recognize nonpresidents, and that the pattern of facial recognition would fit into a serial position curve. Additionally, we identified a group of highly salient presidents—George Washington, Abraham Lincoln, and the presidents who were alive during the subjects' lifetime—and predicted that these *target* presidents would be much easier to recognize than the other presidents. Finally, we predicted that the confidence of hits for presidents would be higher than that of false alarms for nonpresidents, and that the target presidents would be recognized more confidently than the other presidents.

## Experiment 1: Materials & method

### Subjects

We aimed to recruit 200 subjects from Amazon's Mechanical Turk (MTurk) and collected data from 206 adults in the United States (representing 39 states plus Washington, D.C.) who participated by completing an online survey. The data from 13 subjects were excluded from analyses for self-reporting that they used external sources ($N = 11$) or that English was not their primary language ($N = 2$). Among the remaining 193 subjects (87 females and 106 males; $M_{age} = 37.77$, $SD_{age} = 12.52$, age range: 19–73 years), 89 reported being Democratic, 44 Republican, 50 Independent, one other (Libertarian), and nine did not identify with a political party.

The Furman University IRB approved both experiments. When the survey began subjects read a letter containing elements of consent and clicked a button to confirm they wanted to participate (due to the anonymous nature of the study our study was exempted from collecting signatures).

### Materials

We used 132 portraits and photographs of US presidents and other historical figures, including 44 presidents (from George Washington to Donald Trump), 34 vice presidents who did not go on to become presidents, and 54 other lures, such as Civil War generals, famous historical figures, and other politicians. All pictures of the presidents were the official White House portraits collected from The White House Historical Association [15]. The official portraits for the first 42 presidents were painted portraits, whereas the official portraits for Barack Obama and Donald Trump were photographs. Note that Grover Cleveland only appeared once in our

list despite serving two nonconsecutive terms as president. Because Donald Trump was the current president at the time of data collection, we did not include his picture in our analyses.

The nonpresidents used in this study were compiled of the nonpresidents from Roediger and DeSoto [9] and other historical figures; these individuals were gathered from a variety of time periods so that their relative ages and time of service were similar to the presidents. The photographs and portraits were selected from several internet sources (Britannica ImageQuest, History Art and Archives: United States House of Representatives, the White House Historical Association, and Google Images) and appeared similar in style and format to the presidential portraits. All pictures did not contain the person's name, showed more than half of the face of the person, and were not grainy. We adjusted the size on all images to the same resolution on the screen (72 pixels/inch), and we made the height 500 pixels.

### Procedure

Subjects first maximized their web browser and consented to participate before beginning the survey on Qualtrics. They were instructed to complete the study in a quiet environment without engaging in other tasks or using external sources.

After completing the vocabulary subtask of the Shipley General Knowledge Test [16] subjects saw the images of the US presidents and other historical figures presented one at a time in a random order. For each image, subjects indicated whether the person portrayed was a US president by clicking either the "president" or the "not a president" button at the bottom of the screen. Then subjects rated their confidence in their decision on a 100-point visual sliding scale ranging from 0 (*not at all confident*) to 100 (*entirely confident*). After submitting their response, the next portrait appeared. Portrait presentations and confidence ratings were self-paced.

After the recognition task, subject answered questions about their age, gender, handedness, home state, highest level of education, political party association, whether they had lived in the US for at least 5 years, and whether English was their primary language. Additionally, subjects identified which president was the first president they could remember serving during their lifetime and whether they used any external resources (they were told that answering yes would not affect their payment). After completing the study, subjects read a debriefing and instructions for receiving payment.

## Experiment 1: Results & discussion

For all analyses, we used $\alpha = .05$ to indicate statistical significance and Welch two-sample *t*-tests, which adjust for unequal variance between groups. We report one-tailed t-tests when we made directional predictions (as noted in our preregistration) but two-tailed tests yielded the same outcome in every instance. We used Cohen's *d* as our measure of effect size. We used Pearson correlations except when normality assumptions were violated, in which case we used Kendall's Tau [17]. Finally, although we gathered responses to Donald Trump, we did not include this president in our analyses as noted in our preregistration.

### Preregistered analyses

**How well did people recognize the presidents?** As shown in Table 1, subjects were more likely to correctly recognize presidents as presidents than to falsely identify nonpresidents as presidents. In other words, the hit rate for presidents ($M = .58$) was significantly higher than the false alarm rate for nonpresidents ($M = .20$), $M_{\text{diff}} = .38$, 95% CI [.36, .41], $t(192) = 24.25$, $p < .001$, $d = 1.75$. Table 2 shows the proportion of subjects who identified each president as a president. The hit rates varied widely, from a low of James Buchanan at .18 to a high of Clinton

**Table 1. Mean hit and false alarm rates and subjects' mean confidence when responding "president" to an image in Experiment 1.**

|  | Respond "President" [95% CI] | Confidence [95% CI] |
|---|---|---|
| President | .58 [.56, .61] | 81 [79, 83] |
| Nonpresident | .20 [.18, .22] | 63 [60, 65] |

**Table 2. Proportion of subjects who correctly identified presidents as presidents in Experiments 1 and 2, in order of when they served their first term.**

|  | Experiment 1 | | Experiment 2 | |
|---|---|---|---|---|
| Name | Hit Rate | [95% CI] | Hit Rate | [95% CI] |
| George Washington | .93 | [.89, .96] | .99 | [.97, 1.00] |
| John Adams | .43 | [.36, .50] | .80 | [.72, .88] |
| Thomas Jefferson | .77 | [.71, .83] | .94 | [.89, .99] |
| James Madison | .51 | [.44, .58] | .77 | [.69, .85] |
| James Monroe | .26 | [.20, .32] | .61 | [.51, .71] |
| John Quincy Adams | .48 | [.41, .55] | .68 | [.59, .77] |
| Andrew Jackson | .65 | [.59, .72] | .77 | [.69, .85] |
| Martin Van Buren | .52 | [.45, .59] | .63 | [.53, .73] |
| William Henry Harrison | .48 | [.41, .55] | .62 | [.52, .72] |
| John Tyler | .36 | [.29, .43] | .43 | [.33, .53] |
| James K. Polk | .43 | [.36, .50] | .69 | [.60, .78] |
| Zachary Taylor | .28 | [.22, .34] | .43 | [.33, .53] |
| Millard Fillmore | .31 | [.24, .37] | .46 | [.36, .56] |
| Franklin Pierce | .42 | [.34, .48] | .44 | [.34, .54] |
| James Buchanan | .18 | [.13, .24] | .42 | [.32, .52] |
| Abraham Lincoln | .95 | [.92, .98] | 1.00 | [1.00, 1.00] |
| Andrew Johnson | .24 | [.18, .30] | .57 | [.47, .67] |
| Ulysses S. Grant | .37 | [.30, .44] | .53 | [.43, .63] |
| Rutherford B. Hayes | .47 | [.40, .54] | .63 | [.53, .73] |
| James A. Garfield | .50 | [.43, .57] | .70 | [.61, .79] |
| Chester A. Arthur | .42 | [.35, .49] | .45 | [.35, .55] |
| Grover Cleveland | .53 | [.46, .60] | .67 | [.58, .76] |
| Benjamin Harrison | .53 | [.46, .60] | .53 | [.43, .63] |
| William McKinley | .35 | [.28, .41] | .60 | [.50, .70] |
| Theodore Roosevelt | .75 | [.68, .81] | .89 | [.83, .95] |
| William Howard Taft | .77 | [.71, .83] | .88 | [.82, .94] |
| Woodrow Wilson | .50 | [.43, .57] | .80 | [.72, .88] |
| Warren G. Harding | .31 | [.24, .38] | .40 | [.30, .50] |
| Calvin Coolidge | .21 | [.15, .27] | .53 | [.43, .63] |
| Herbert Hoover | .46 | [.39, .53] | .65 | [.55, .75] |
| Franklin D. Roosevelt | .59 | [.52, .66] | .81 | [.73, .89] |
| Harry S. Truman | .73 | [.67, .79] | .87 | [.80, .94] |
| Dwight D. Eisenhower | .74 | [.67, .80] | .84 | [.77, .91] |
| John F. Kennedy | .88 | [.83, .92] | .97 | [.94, 1.00] |
| Lyndon B. Johnson | .82 | [.76, .87] | .92 | [.87, .97] |
| Richard M. Nixon | .92 | [.88, .96] | .90 | [.84, .96] |
| Gerald R. Ford | .52 | [.45, .59] | .55 | [.45, .65] |

*(Continued)*

**Table 2.** (Continued)

| | Experiment 1 | | Experiment 2 | |
|---|---|---|---|---|
| **Name** | **Hit Rate** | **[95% CI]** | **Hit Rate** | **[95% CI]** |
| Jimmy Carter | .83 | [.78, .89] | .81 | [.73, .89] |
| Ronald Reagan | .94 | [.90, .97] | .98 | [.95, 1.00] |
| George Bush | .90 | [.86, .94] | .97 | [.94, 1.00] |
| William J. Clinton | .98 | [.97, 1.00] | .78 | [.70, .86] |
| George W. Bush | .96 | [.93, .99] | .97 | [.94, 1.00] |
| Barack Obama | .98 | [.96, 1.00] | 1.00 | [1.00, 1.00] |

and Obama at .98. Likewise, Table 3 shows the false alarm rates, or the proportion of subjects who falsely identified nonpresidents as presidents. Again, there was a wide range, from the low of Hillary Clinton (.03) and Colin Powell (.06) to the high of Alexander Hamilton (.66).

**How confident were people in their presidential identifications?** Table 1 shows that as expected, subjects were more confident when correctly identifying presidents ($M = 81$) than when falsely identifying nonpresidents as presidents ($M = 63$), $M_{\text{diff}} = 18.05$, 95% CI = [16, 20], $t(189) = 20.43$, $p < .001$, $d = 1.48$. This pattern of confidence ratings replicated prior work in presidential name recognition [9].

**Did people recognize our "target" presidents exceptionally well?** We expected that subjects would have excellent recognition for the presidents who were alive during the subjects' lifetime, which was deducted from each subject's reported age and the starting term of each president. On average, 5.79 presidents have served during each subject's lifetime, with a range from three to eleven. In general, the pattern of recognition fits a serial position curve (Fig 1). Subjects recognized the first president (George Washington), the most recent presidents (presidents who were alive during the subjects' lifetime), and Abraham Lincoln, better than the other presidents. Subjects recognized these target presidents ($M = .94$) more frequently than the non-target presidents ($M = .50$), $M_{\text{diff}} = .44$, 95% CI = [.41, .47], $t(192) = 30.36$, $p < .001$, $d = 2.19$. This pattern is generally consistent with previous research on presidential name recognition [9], although our face recognition data show a lower recognition rate. Furthermore, subjects were more confident in correctly recognizing the aforementioned selected presidents ($M = 95$) than the other presidents ($M = 68$), $M_{\text{diff}} = 27$, 95% CI = [25, 30], $t(192) = 21.43$, $p < .001$, $d = 1.54$.

## Exploratory analyses

**Was correct recognition correlated with vocabulary?** Subjects who performed better on the Shipley vocabulary task also performed better on the presidential recognition test. The proportion of correct responses (hits plus correct rejections divided by the total number of items) on the recognition task correlated positively with scores on the Shipley vocabulary task, $r_\tau = .39$, $p < .001$. This correlation suggests that subjects with higher verbal intelligence tend to better recognize faces of the US presidents.

**How did facial recognition compare to name recognition?** There was a strong correlation between the hit rate in the current study and that in Roediger and DeSoto [9], $r_\tau = .50$, $p < .001$, 95% CI [.35, .65], indicating that knowledge of the names of the presidents is similar to knowledge for the faces of the presidents.

**What was the relationship between confidence and responding "president"?** We also examined the relationship between responding "President" and confidence in those recognition decisions as shown in Fig 2, which displays the proportion of subjects who responded

**Table 3. Proportion of subjects who incorrectly identified nonpresidents as presidents in Experiments 1 and 2.**

|  | Experiment 1 | | Experiment 2 | |
|---|---|---|---|---|
| Name | False Alarm Rate | [95% CI] | False Alarm Rate | [95%CI] |
| Aaron Burr | .25 | [.19, .32] | .30 | [.21, .39] |
| Adam Smith | .18 | [.12, .23] | .32 | [.23, .41] |
| Adlai Stevenson | .26 | [.20, .32] | .14 | [.07, .21] |
| Al Gore | .13 | [.08, .18] | .10 | [.04, .16] |
| Alben W. Barkley | .22 | [.16, .28] | .23 | [.15, .31] |
| Alexander Bell | .08 | [.04, .12] | .19 | [.11, .27] |
| Alexander Hamilton | .66 | [.59, .73] | .53 | [.43, .63] |
| Alfred Landon | .13 | [.09, .18] | .16 | [.09, .23] |
| Alfred Smith | .15 | [.09, .20] | .17 | [.10, .24] |
| Andrew Carnegie | .38 | [.31, .45] | .26 | [.17, .35] |
| Benjamin Franklin | .21 | [.15, .26] | .36 | [.26, .46] |
| Bill Archer | .19 | [.13, .24] | .30 | [.21, .39] |
| Carl Albert | .17 | [.12, .22] | .16 | [.09, .23] |
| Charles Curtis | .17 | [.11, .22] | .17 | [.10, .24] |
| Charles G. Dawes | .07 | [.03, .10] | .19 | [.11, .27] |
| Charles W. Fairbanks | .23 | [.17, .29] | .24 | [.16, .33] |
| Colin Powell | .06 | [.02, .09] | .04 | [.00, .08] |
| Cordell Hull | .16 | [.10, .21] | .16 | [.09, .23] |
| Dan Burton | .15 | [.09, .20] | .23 | [.15, .31] |
| Dan Quayle | .17 | [.11, .22] | .23 | [.15, .31] |
| Daniel D. Tompkins | .21 | [.15, .27] | .21 | [.13, .29] |
| Dick Cheney | .10 | [.06, .14] | .11 | [.05, .17] |
| Elbridge Gerry | .44 | [.37, .51] | .30 | [.21, .39] |
| Felix Edward Hebert | .35 | [.28, .41] | .28 | [.19, .37] |
| Francis Parkman | .10 | [.06, .14] | .15 | [.08, .22] |
| Frederick Conrad | .20 | [.14, .26] | .22 | [.14, .30] |
| Garret Hobart | .34 | [.27, .40] | .31 | [.22, .40] |
| George Clinton | .22 | [.16, .28] | .24 | [.15, .33] |
| George H. Mahon | .25 | [.19, .32] | .27 | [.18, .36] |
| George M. Dallas | .25 | [.19, .31] | .13 | [.06, .20] |
| Gideon Welles | .09 | [.05, .13] | .11 | [.05, .17] |
| Hannibal Hamlin | .17 | [.11, .22] | .11 | [.05, .17] |
| Harold Ickes | .22 | [.16, .28] | .17 | [.10, .24] |
| Harold Stassen | .09 | [.05, .13] | .12 | [.06, .18] |
| Henry A. Wallace | .20 | [.14, .25] | .44 | [.34, .54] |
| Henry Ford | .19 | [.14, .25] | .30 | [.21, .39] |
| Henry Stimson | .12 | [.08, .17] | .08 | [.03, .13] |
| Henry Wilson | .13 | [.08, .18] | .22 | [.14, .30] |
| Hillary Clinton | .03 | [.01, .06] | .01 | [.00, .03] |
| Hubert Humphrey | .32 | [.25, .39] | .56 | [.46, .66] |
| Hugo Black | .13 | [.08, .18] | .06 | [.01, .11] |
| Jack B. Brooks | .11 | [.06, .15] | .15 | [.08, .22] |
| James S. Sherman | .17 | [.11, .22] | .33 | [.24, .42] |
| Joe Biden | .13 | [.09, .18] | .18 | [.10, .26] |
| John Blatnik | .34 | [.27, .41] | .22 | [.14, .30] |
| John C. Breckinridge | .10 | [.06, .15] | .12 | [.06, .18] |

*(Continued)*

**Table 3.** (Continued)

| Name | Experiment 1 | | Experiment 2 | |
|---|---|---|---|---|
| | False Alarm Rate | [95% CI] | False Alarm Rate | [95%CI] |
| John C. Calhoun | .41 | [.34, .48] | .48 | [.38, .58] |
| John Hancock | .31 | [.24, .37] | .29 | [.20, .38] |
| John Jay | .26 | [.20, .32] | .31 | [.22, .40] |
| John N. Garner | .15 | [.10, .20] | .16 | [.09, .23] |
| Joseph William Martin | .12 | [.07, .17] | .18 | [.10, .26] |
| Leslie C. Arends | .10 | [.06, .15] | .12 | [.06, .18] |
| Levi P. Morton | .14 | [.09, .19] | .15 | [.08, .22] |
| Mike Pence | .12 | [.07, .17] | .17 | [.10, .24] |
| Nathaniel Greene | .16 | [.11, .21] | .20 | [.12, .28] |
| Nelson Rockefeller | .15 | [.10, .20] | .20 | [.12, .28] |
| Oliver Holmes | .09 | [.05, .13] | .05 | [.01, .09] |
| Orville Wright | .11 | [.06, .15] | .11 | [.05, .17] |
| Patrick Henry | .18 | [.12, .23] | .32 | [.23, .41] |
| Paul Revere | .14 | [.09, .19] | .14 | [.07, .21] |
| Richard K Armey | .22 | [.16, .28] | .27 | [.18, .36] |
| Richard M. Johnson | .48 | [.41, .55] | .41 | [.31, .51] |
| Robert E. Lee | .19 | [.14, .25] | .25 | [.16, .34] |
| Samuel Adams | .33 | [.26, .40] | .42 | [.32, .52] |
| Samuel Morse | .31 | [.24, .37] | .26 | [.17, .35] |
| Schuyler Colfax | .16 | [.10, .21] | .13 | [.06, .20] |
| Spiro Agnew | .39 | [.32, .46] | .19 | [.11, .27] |
| Stephen Austin | .38 | [.31, .45] | .18 | [.10, .26] |
| Stephen Douglas | .20 | [.14, .26] | .30 | [.21, .39] |
| Stonewall Jackson | .11 | [.07, .16] | .18 | [.10, .26] |
| Thomas A. Hendricks | .26 | [.20, .32] | .28 | [.19, .37] |
| Thomas Austin | .11 | [.07, .16] | .04 | [.00, .08] |
| Thomas Edison | .21 | [.15, .27] | .28 | [.19, .37] |
| Thomas R. Marshall | .28 | [.22, .35] | .40 | [.30, .50] |
| Timothy Pickering | .28 | [.22, .35] | .22 | [.14, .30] |
| Tom Bliley | .10 | [.06, .14] | .10 | [.04, .16] |
| W.R. Poage | .17 | [.12, .22] | .28 | [.19, .37] |
| Walter Mondale | .10 | [.06, .15] | .08 | [.03, .13] |
| Wendell Wilkie | .09 | [.05, .13] | .09 | [.03, .15] |
| William A. Wheeler | .10 | [.06, .15] | .14 | [.07, .21] |
| William Bryan | .10 | [.06, .14] | .25 | [.16, .34] |
| William Clark | .39 | [.32, .46] | .46 | [.36, .56] |
| William R. King | .29 | [.23, .35] | .19 | [.11, .27] |
| William Seward | .26 | [.20, .32] | .21 | [.13, .29] |
| William Tweed | .19 | [.14, .25] | .26 | [.17, .35] |
| Winston Churchill | .27 | [.21, .33] | .26 | [.17, .35] |
| Wright Patman | .20 | [.14, .25] | .15 | [.08, .22] |
| Zebulon Vance | .17 | [.12, .22] | .11 | [.05, .17] |

"president" to each of our images along with the confidence in those decisions. There was a strong positive correlation, $r(129) = .77$, $p < .001$, 95% CI = [.68, .83], indicating that the faces received higher confidence ratings when the judgment of "president" was assigned.

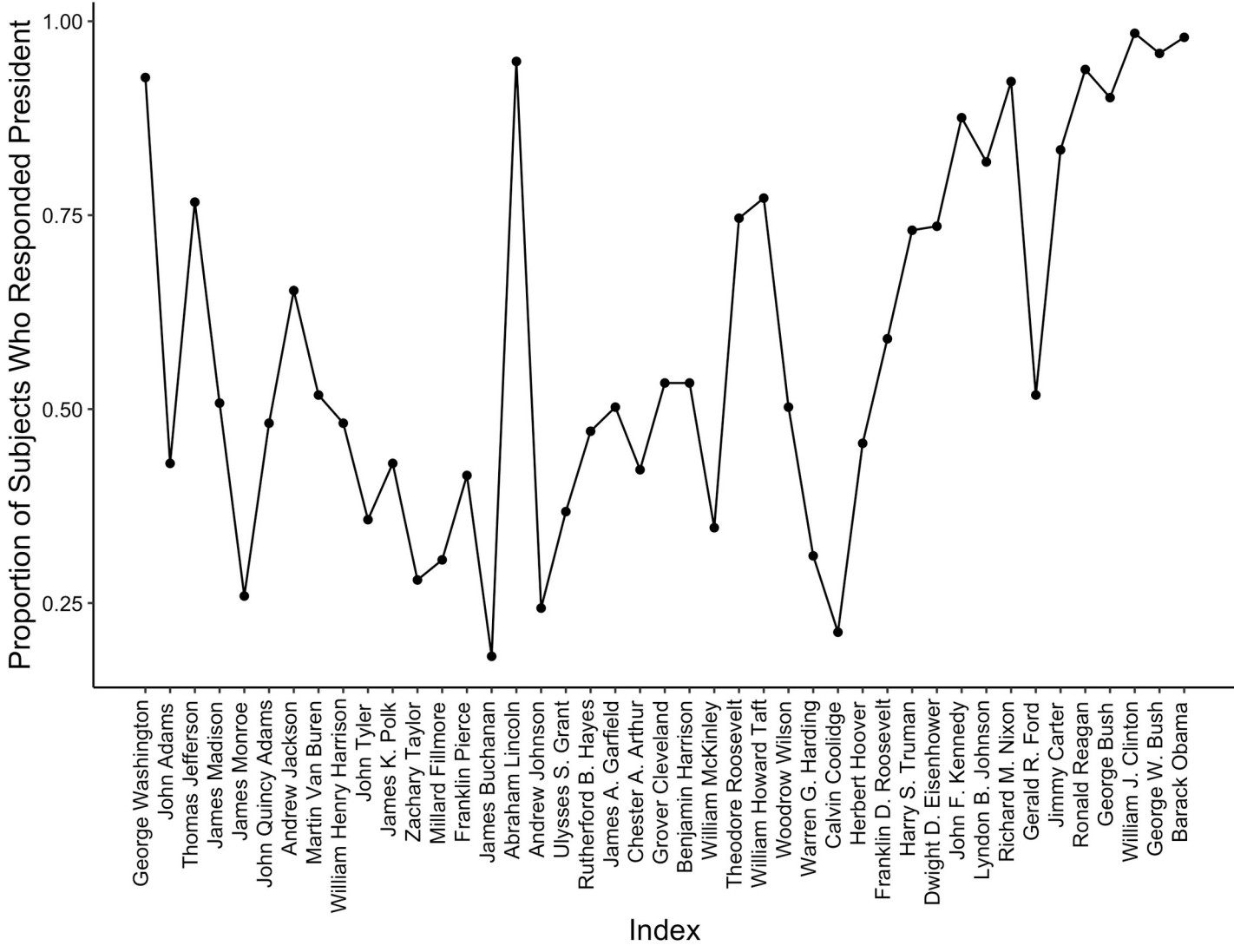

**Fig 1. Serial position curve showing the proportion of subjects who responded "president" to each president's image in Experiment 1.**

*Did people falsely identify Alexander Hamilton as a president*? Roediger and DeSoto [9] found that Alexander Hamilton was an outlier: many people identified him as a president with high confidence in response to his name. In the current experiment, Hamilton had a mean false alarm rate of .66 and a mean confidence rating of 66.6, suggesting that Americans falsely recognize not only his name but also his face at a fairly high rate; in fact, this face false alarm rate was higher than the face hit rate for 27 out of the 43 (61%) presidents in our study (Hamilton is the clear nonpresident outlier in Fig 2).

*Does appearing on a bill or coin increase correct and false recognition rates*? One question is whether a face is more readily called a president if that individual appears on a US bill or coin. Currently, seven presidents (Washington, Jefferson, Jackson, Lincoln, Grant, F. D. Roosevelt, and Kennedy) and two nonpresidents (Hamilton and Franklin) appear on US currency. Table 4 presents the mean presidential response rate and confidence for the presidents and nonpresidents as a function of whether they appear on US currency. As seen in the table, individuals depicted on currency had higher average "president" response rates for both presidents

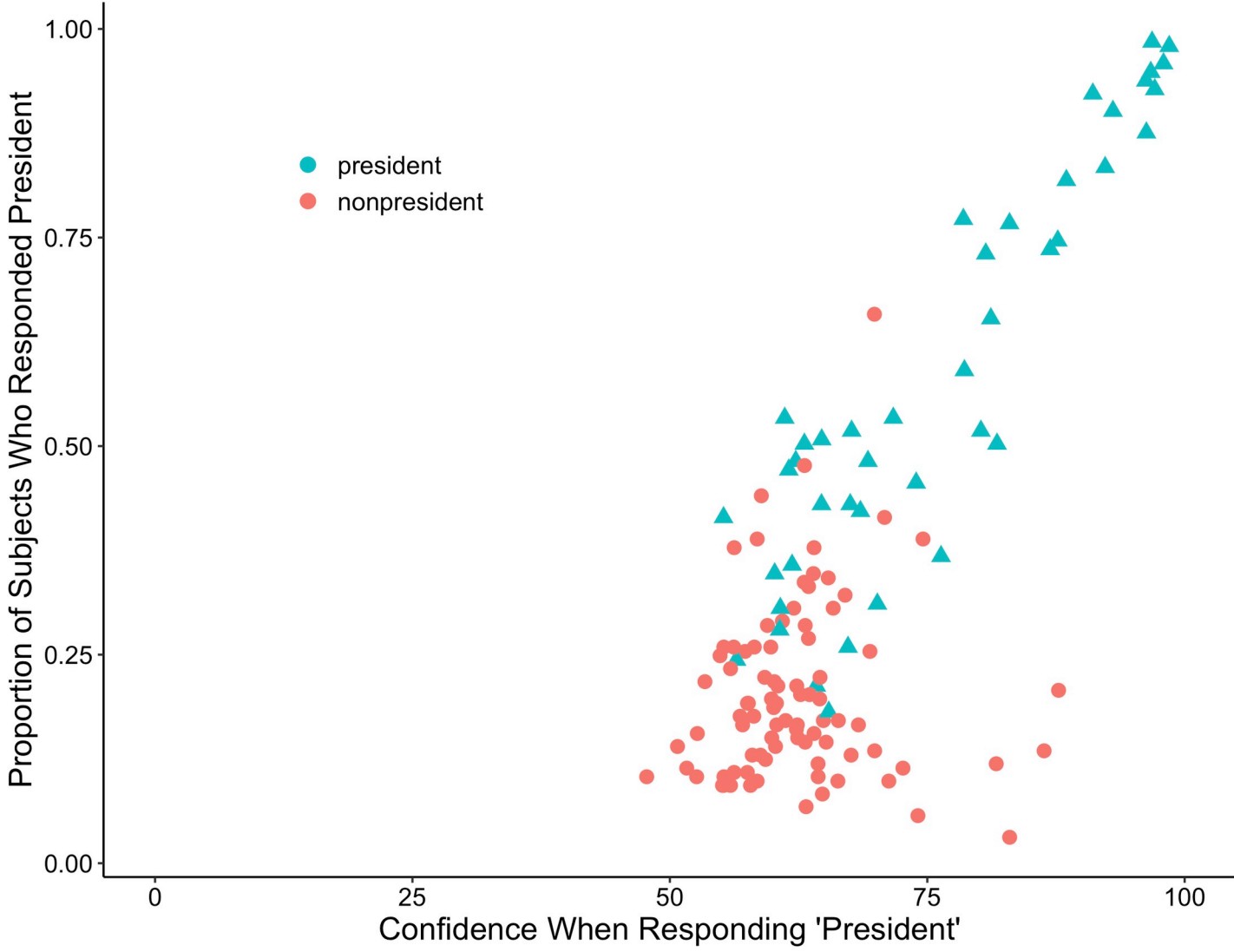

**Fig 2. Scatterplot showing the association between the proportion of subjects who identified each image as a president and the mean confidence rating for that choice.** Each data point represents a particular portrait, with triangles representing presidents and circles representing nonpresidents. Alexander Hamilton was the nonpresident outlier.

(the hit rate) and nonpresidents (the false alarm rate). Confidence was similar for presidents regardless of whether they appeared on currency or not, but false alarms were made with more confidence for the two nonpresidents who appeared on currency. Because there are so few

**Table 4. Mean hit and false alarm rates and subjects' mean confidence when responding "president" to a face in Experiment 1 as a function of whether the individual appears on US currency.**

| | Respond "President" | | Confidence | |
|---|---|---|---|---|
| | **On Currency** | **Not on Currency** | **On Currency** | **Not on Currency** |
| President | .73 (.21) | .56 (.24) | 61 (5) | 61 (5) |
| Nonpresident | .43 (.32) | .20 (.10) | 79 (13) | 62 (7) |

Note: *SD* appears in parentheses.

presidents on bills or coins (and even fewer nonpresidents), we will limit our examination to descriptive statistics only and avoid making inferential claims.

## Experiment 2

Experiment 1 examined Americans' facial recognition rates of the US presidents and showed a serial position curve that showed excellent memory for Washington, Lincoln, and the more recent presidents. However, the conclusions about the comparisons are limited because the name and facial recognition data were collected from two different samples at two different times. Thus, Experiment 2 directly compared memory for the presidents' faces with memory for their names in a single experiment. To do so, we added a within-subjects manipulation in Experiment 2, so that subjects saw a mixed list of faces and names of presidents and nonpresidents. We anticipated that subjects would generally be less accurate at identifying faces than names, that there would be a strong correlation between the hit rates for faces and for names, that confidence for hits would be higher than confidence for false alarms, and that subjects would be similarly confident in their judgments for faces and names. In addition, we expected that, similar to Experiment 1, subjects would have particularly good memory for Washington, Lincoln, and presidents who were alive during their lifetime. Finally, we expected that overall recognition accuracy would be related positively to general knowledge as measured by vocabulary.

## Experiment 2: Materials & method

Experiment 2 was preregistered on the open science framework at OSF.IO/6TWDH. We report all conditions, measures, analyses, and describe how we determined our sample size.

### Subjects

One hundred ten undergraduates participated for class credit. Our goal for the sample size was to collect data from as many subjects as possible before the end of the semester, with the minimum being 100 subjects. The exclusion criteria were similar to those in Experiment 1. In the end, we excluded data from 10 subjects from all analyses: seven for reporting using external resources and three for reporting that English was not their first language. Among the final 100 subjects (61 females and 39 males, $M_{age}$ = 19.14, $SD_{age}$ = 1.06, age range: 18–22 years), 22 reported being Democratic, 40 Republican, 10 Independent, two other, and 26 did not identify with any political party. These subjects came from 23 different US states.

### Materials and procedure

The materials were identical to Experiment 1 except that subjects were asked to identify presidents in response to 66 faces and 66 names. In most cases, presidents were referred to by their full names (e.g., William J. Clinton rather than Bill Clinton), but some presidents were referred to by their more commonly known nicknames (e.g., Jimmy Carter rather than James Earl Carter Jr.). The presidential names were presented as listed on The White House Historical Association site [15]. We created two counterbalanced versions of our stimuli list from Experiment 1. In the first version, we assigned the odd-numbered presidents and nonpresidents to be faces and the even-numbered ones to be names; whereas in the second version, we assigned the even numbers to be faces and the odd numbers to be names. Subjects were randomly assigned to either List 1 or List 2. The rest of the procedure was the same as in Experiment 1.

## Experiment 2: Results & discussion

As in Experiment 1, although we included Donald Trump as a target president, we did not include responses to this item in our analyses as he was the current president when data was collected. We ran one-tailed *t*-tests when we made directional *a priori* predictions, but two-tailed *t*-tests yielded similar conclusions in all cases.

### Preregistered analyses

**How well did people recognize the names and faces of the presidents?** First, as seen in Table 5, we examined the overall hit rate for the presidents and false alarm rate for the nonpresidents as a function of whether subjects responded to names or faces. As predicted, subjects were less likely to judge the faces of presidents ($M = .70$) as presidents than the names of presidents as presidents ($M = .73$), $M_{diff} = .03$, 95% CI = [.01, .06], $t(99) = -2.57$, $p = .006$, $d = 0.26$. Fig 3 displays the hit rate plotted by ordinal position, showing a serial position curve. Subjects were also more likely to falsely identify the faces of nonpresidents as presidents ($M = .30$) than the names of nonpresidents as presidents ($M = .14$, $M_{diff} = .16$, 95% CI = [.13, .18]), $t(99) = 11.40$, $p < .001$, $d = 1.14$. Together, these results suggest that Americans have better memory for the names—rather than faces—of the presidents. This pattern occurred despite a medium to strong correlation between the hit rate for names and faces, $r_\tau = .42$, $p < .001$, 95% CI = [.30, .54].

**How confident were people in their presidential identifications?** Our second set of analyses examined the overall confidence ratings and the confidence ratings as a function of whether subjects responded to names or faces. Because not all subjects had relevant data points (e.g., they might not have committed a false alarm in response to a name, so there are no confidence ratings for that cell), the degrees of freedom differed for some analyses. Table 6 shows the subjects' mean confidence ratings when responding "president" to both faces and names (i.e., the confidence for hits and false alarms as a function of responding to faces and names). Overall, subjects were confident when they correctly recognized a president ($M = 86$) and less confident when they falsely identified a nonpresident as a president ($M = 65$, $M_{diff} = 21$, 95% CI = [19, 23]), $t(99) = 18.90$, $p < .001$, $d = 1.90$.

In examining the confidence ratings as a function of rating faces and names, we found, as predicted, that subjects were similarly confident when they correctly recognized the faces ($M = 86$) compared to when they correctly recognized the names of the presidents ($M = 86$), $M_{diff} = 0.63$, 95% CI = [−2, 1], $t(99) = -0.74$, $p = .463$, $d = 0.07$. Curiously, however, we observed a different pattern for false alarms: subjects were more confident when they falsely identified the names ($M = 70$) than when they falsely identified the faces of nonpresidents as those of presidents, ($M = 62$), $M_{diff} = 8$, 95% CI = [5, 12], $t(90) = 4.88$, $p < .001$, $d = 0.51$. Overall, this latter finding suggests that while Americans were less likely to false alarm to a name than to a face, when they did false alarm, they were more confident for names than faces.

### Did people recognize our "target" presidents exceptionally well?

Our third analysis examined whether subjects had better memory for our target presidents (Washington, Lincoln, and the presidents who were alive during the subjects' lifetime) than

**Table 5. Mean hit and false alarm rates for responding "president" to a name versus to an image in Experiment 2.**

|  | Stimulus Type | |
|---|---|---|
|  | **Face [95% CI]** | **Name [95% CI]** |
| President | .70 [.68, .73] | .73 [.71, .76] |
| Nonpresident | .30 [.27, .32] | .14 [.12, .16] |

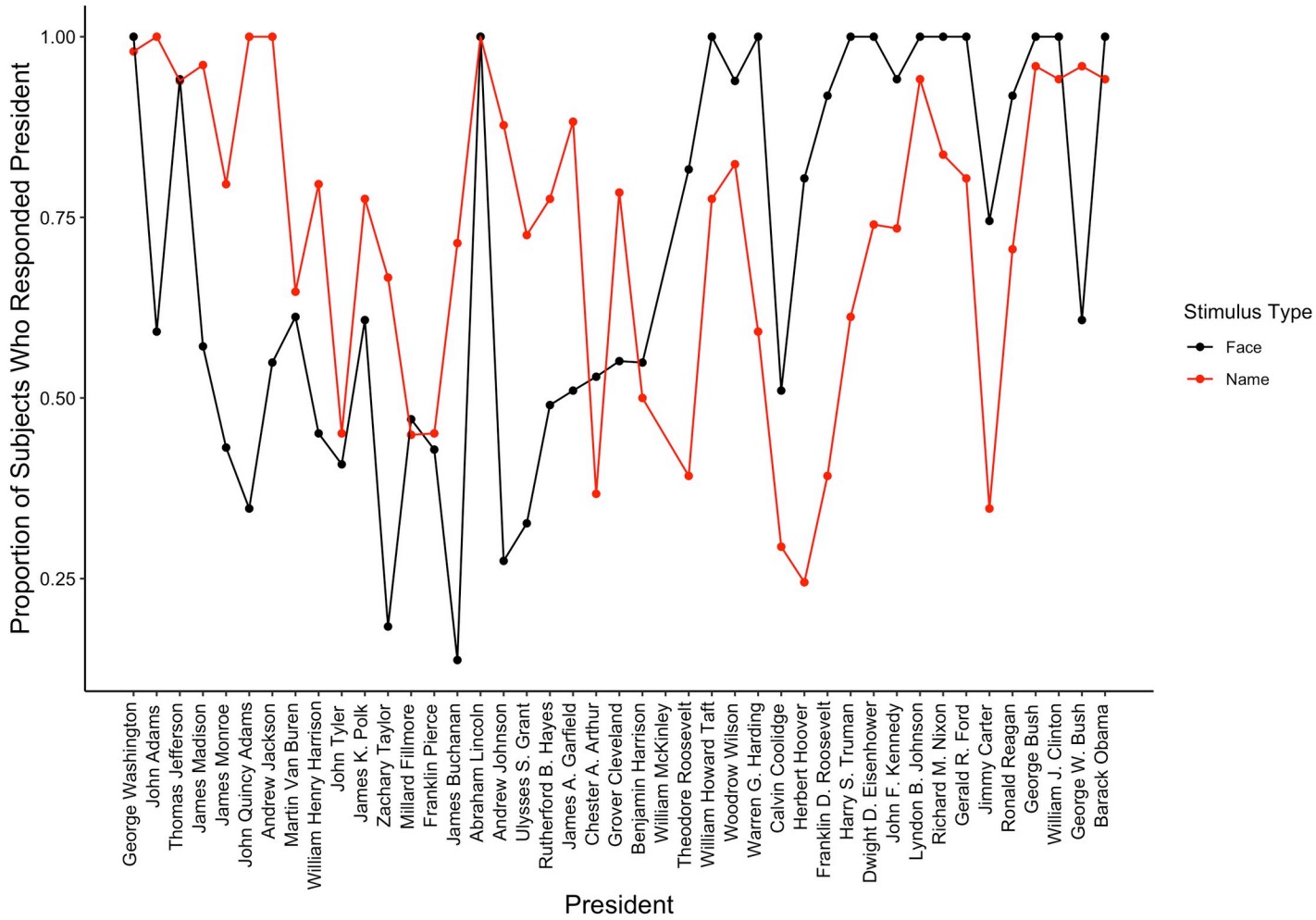

**Fig 3. Proportion of subjects who responded "president" in response to seeing a president's name or face.**

the other presidents. As expected, the hit rate for the target presidents ($M = .96$) was much higher than that for the other presidents ($M = .69$), $M_{diff} = .28$, 95% CI = [.25, .30], $t(97) = 23.04$, $p < .001$, $d = 2.03$. Indeed, only three subjects in our sample did not have a perfect hit rate for the selected presidents.

**Was correct recognition correlated with vocabulary?** Finally, similar to what we found in Experiment 1, the overall correct recognition rate (responding "president" to presidents and "nonpresident" to nonpresidents) was positively associated with scores on the Shipley task, $r(98) = .24$, $p = .018$, 95% CI = [.04, .41], suggesting that subjects who had larger vocabulary did better on the president recognition task.

**Table 6. Mean confidence ratings for correctly responding "president" to a name versus a face in Experiment 2.**

|  | Stimulus Type | | |
|---|---|---|---|
|  | **Face** | **Name** | **Average** |
| President | 86 [84, 88] | 86 [83, 88] | 86 [84, 88] |
| Nonpresident | 62 [59, 66] | 70 [66, 74] | 65 [61, 68] |

## Exploratory analyses

**What were the overall hit and false alarm rates?**  We examined the overall hit and false alarm rates collapsed across responses for both faces and names. Overall, subjects were accurate in identifying presidents ($M$ = .72, 95% CI = [.66, .74]) and were unlikely to falsely identify nonpresidents as presidents ($M$ = .22, 95% CI = [.20, .24]. Tables 3 and 4 display the recognition and false recognition rates for presidents and nonpresidents. As in Experiment 1 there was a wide range for both presidents (ranging from a low of .30 for Warren G. Harding to a high of 1.00 for Lincoln and Obama) and nonpresidents (ranging from a low of .00 for Colin Powell to a high of .56 for Hubert Humphrey).

**What was the relationship between confidence and responding "president"?**  As shown in the two panels in Fig 4, there was a strong correlation between responding "president" to an individual and the confidence rating for that response, $r_\tau$ (129) = .58, $p < .001$, 95% CI = [.47, .69]. The correlation was similar regardless of whether subjects were responding to faces, $r_\tau$ (129) = .50, $p < .001$, 95% CI [.39, .61], or to names, $r_\tau$ (129) = .57, $p < .001$, 95% CI [.47, .66]. Both of these correlations were much weaker than the one found in Roediger and DeSoto [9] where the correlation was .95, 95% CI = [.93, .97]. One possibility for this discrepancy is that the samples in the two studies differed; subjects in this experiment were college students, whereas the sample in the Roediger and DeSoto study was drawn from an older MTurk population. Another possibility is that in this experiment, the correlation for names might be lower due to high confidence false alarms.

**Which presidents were the most difficult to recognize?**  As shown in Table 2, we found that seven presidents were recognized less than 50% of the time, collapsing across names and faces: Warren G. Harding, James Buchanan, John Tyler, Zachary Tyler, Franklin Pierce, Chester A. Arthur, and Millard Fillmore.

We also examined whether recognition rates were different for faces and names. When making decisions based on names, Calvin Coolidge was the least recognized president (hit rate = .24); but when making decisions based on faces, James Buchanan was the least recognized president (hit rate = .14).

Some presidents may be recognized based on their appearance despite people not recognizing their names. Calvin Coolidge, for example, had a recognition rate of .24 when subjects saw his name but a recognition rate of .80 when they saw his picture. Similarly, Herbert Hoover

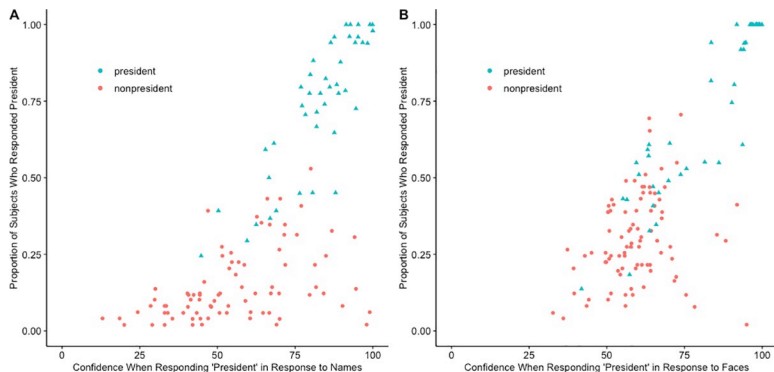

**Fig 4. Scatterplots showing the association between the proportion of subjects who identified each individual as a president and the mean confidence rating for that choice.** Each data point represents an individual, with triangles representing presidents and circles representing nonpresidents. Panel A represents names and Panel B represents faces. Note that six nonpresidents in Panel A and one nonpresident in Panel B are omitted because no subject false alarmed to those items.

**Table 7. Mean hit and false alarm rates and subjects' mean confidence when responding "president" to a face or name in Experiment 2 as a function of whether the individual appears on US currency.**

| | Respond "President" | | Confidence | |
|---|---|---|---|---|
| | On Currency | Not on Currency | On Currency | Not on Currency |
| Faces | | | | |
| President | .83 (.28) | .68 (.27) | 89 (13) | 78 (17) |
| Nonpresident | .56 (.21) | .29 (.14) | 83 (13) | 59 (11) |
| Names | | | | |
| President | .89 (.15) | .70 (.23) | 93 (11) | 80 (13) |
| Nonpresident | .33 (.03) | .13 (.12) | 83 (16) | 55 (19) |

Note: *SD* appears in parentheses.

had a name hit rate of only .39 but face hit rate as high as .92. In contrast, 100% of our sample recognized the name "John Quincy Adams" as the name of a president, but only 35% recognized his face. Likewise, Andrew Johnson (name hit rate = .87, face hit rate = .27) and James Buchanan (name hit rate = .71, face hit rate = .14) were also frequently identified by their names but were difficult to recognize by their faces. The different recognition patterns for faces and names could be because some presidents have a face that looks more "presidential" than others.

**Which nonpresidents had high false alarm rates?** Collapsing across names and faces, Hubert Humphrey, Samuel Adams, Henry A. Wallace, William Clark, John C. Calhoun, and Alexander Hamilton were all identified as presidents at a higher rate than the least frequently recognized president, Warren G. Harding (hit rate = .40). Humphrey (the vice president to President Lyndon B. Johnson) was the nonpresident most likely to be falsely identified as a president (false alarm rate = .56), followed by Hamilton (false alarm rate = .53), who was not far behind. In terms of names, Samuel Adams had the highest false alarm rate at .53 (possibly because of increase familiarity of his name associated with the Boston-based beer company); and, in terms of faces, Hamilton had the highest false alarm rate at .71.

Robert K. Armey was a nonpresident who had a low name false alarm rate of .04, but a face false alarm rate of just above chance, .51. This pattern might have occurred because subjects recognized Armey's face (he was the majority leader in congress during the 90s), but not his name, as he went by Dick Armey. The opposite case was Patrick Henry, a leading figure in the American Revolution who served as governor of Virginia twice but never served as a president. He had a name false alarm rate of .43 but a face false alarm rate of only .20.

**Do people still think Alexander Hamilton was a president?** Similar to observations from prior work, subjects in the current experiment false alarmed to Hamilton at a high rate (*M* = .53), with somewhat high confidence (*M* = 76). The high false alarm rate appeared to be driven by the high false alarm rate to Hamilton's face (*M* = .71) rather than his name (*M* = .35).

*Does appearing on a bill or coin increase correct and false recognition rates*? As in Experiment 1, we examined whether appearing on US currency increases the chance that Americans think that person was a president. As seen in Table 7, individuals depicted on currency had higher average "president" response rates and confidence ratings for both presidents (the hit rate) and nonpresidents (the false alarm rate). Notably, this occurred for both faces and names.

## General discussion

The current study examined how well Americans recognized the faces and names of the US presidents. Overall, our results are consistent with past research in showing that Americans

have reasonably good memory for their presidents and that facial recognition displays similar patterns to name recognition—namely, a serial position curve with a spike in recognition for Lincoln [5, 9]. However, there are also some differences between facial and name recognition.

Experiment 1 demonstrated that the facial recognition rates were generally lower than the name recognition rates reported in prior work [9]. Experiment 2 directly compared name and facial recognition and confirmed our hypotheses regarding the name versus facial recognition patterns and confidence ratings: Subjects were less likely to correctly recognize faces of presidents or to falsely identify nonpresidents as presidents when they were presented with images instead of names. The confidence ratings were higher when subjects falsely recognized a name than a face but did not differ for correct recognitions across the two modes of presentation.

The poorer recognition accuracy for the faces of the presidents we found in this study contradicts predictions from the massive literature on the picture superiority effect showing that memory for images is consistently better than memory for words [18]. Why is this the case? We propose two non-mutually exclusive possibilities. One is that, as addressed before, in both historical documents and everyday conversations, past presidents are typically referred to only by their names rather than their images. The lack of repeated exposure to pictures during encoding then leads to an overall lower recognition accuracy [19].

A second explanation is that whereas names essentially have only one form, visual representations of a person can take many forms. One reason that images are normally remembered better than words comes from dual-coding theory, which states that pictures enable encoding of the stimuli as both images and verbal traces. In prior research [10], for example, dual-coding of images facilitated memory as both the pictorial information and the verbal labels that can be used for verbal recall at the test. For the faces used in our study, however, dual-coding of one portrait might not necessarily assist with the recognition of a different portrait of the same person as one's physical appearance may vary in different pictures and photographs. In addition, different sources may include different portraits yet the same names of the US presidents, which leads to unbalanced repetitions of verbal versus pictorial representations of the presidents in Americans' daily encounters and memory. Therefore, prior exposure to pictures of some past presidents might have facilitated only the recognition of the names but not the faces of the same presidents.

Our confidence rating data also presented a somewhat unexpected pattern. The generally higher confidence ratings for correct recognition across conditions were consistent with the results in Roediger and DeSoto [9]. However, the direction in which the confidence ratings for false alarms differed for names and faces was inconsistent with past work suggesting that photos increase feelings of "truthiness" [20]. Images provide semantically rich context that biases people's judgement of the truthiness of uncertain claims by inducing illusions of familiarity, thereby increasing the likelihood for people to believe the claims are true [20]. Applied to the current study, this theory suggests that Americans should be more confident when false alarming to faces than to names, yet our results suggest the opposite. One possibility for this inconsistency is that the illusion of familiarity with the nonpresidents introduced by portraits is not comparable to that by the textual sources that Americans normally encounter. Americans might have been frequently exposed to the names of the nonpresidents included in this study without pictorial references as they read the newspaper and history textbooks. Such exposure may be so frequent that the truthiness effect it produces exceeds that produced by the images they saw during the recognition task, thereby leading to a higher confidence rating for false alarms in the name condition than in the face condition.

No study about recognizing the presidents would be complete without a discussion of Alexander Hamilton. Hamilton has a consistently high overall false alarm rate with somewhat high confidence rating across studies, likely due to the important roles he played in American

history: he co-authored *The Federalist Papers*, dueled with Vice President Aaron Burr, served as Secretary of the Treasury, and was a founding father of the United States [9]. While past research showed that Americans frequently thought Hamilton was a president (71% in Roediger & DeSoto [9]), our study showed a slightly different pattern of recognition: across both experiments, Hamilton had a high face false alarm rate (71%) but a lower name false alarm rate (35%). However, as the musical *Hamilton* became increasingly popular and started its yearly North American tours in 2017, people may have learned more about Hamilton and now realize that he never served as a president. Thus, when responding to Hamilton's name, Americans may now be able to specifically recollect Hamilton's role in the founding of the country (an important founding figure, but not a president) to counteract the high familiarity associated with his name [21]. While the musical dissociates Hamilton's name with the presidency, as Hamilton is played by actors, his face is *not* dissociated with the presidency. Compounding this issue is that Hamilton's face appears on the $10 bill, through the repeated exposure to which Americans still have a high visual fluency for Hamilton's face–indeed the false alarm rate is much higher for Hamilton and Benjamin Franklin (the two nonpresidents who appear on currency) compared to the other nonpresidents. Together, this high visual fluency and association may lead to Americans' false impression that his face is that of a president.

The current study not only confirms that facial recognition for the US president fits into the typical pattern found in verbal collective memory research but also demonstrates that names may not represent the only way Americans recognize their presidents. Previous research has shown that facial and name recognition may be two distinct processes, with some overlap, in person identification. Research [22] indicates that while there are overlapping brain regions associated with name and facial recognition, there are also distinct areas of activation when processing and recognizing names versus faces. Our results are consistent with this view: there is some correlation between the recognition of names and faces that indicates some degree of a modality-agnostic person identification system, but there are also differences in the recognition rates, suggesting that differences also likely exist in some early pre-semantic perceptual processing for names and faces.

Finally, the current work extends the scope of quantitative research on collective memory. To date, most quantitative collective memory work in psychology has focused on verbal memories (for a few exceptions, see [23–25]). Collective memory scholars outside of psychology, however, have long argued that collective memories are often created, maintained, and remembered in nonverbal ways, such as through commemorative statues, museums, and public memorials. Connerton [26], for instance, observes that collective memories are at times procedural or nondeclarative memories—that remembering occurs in the context of a ceremony or ritual. One example here is the Passover Seder, which helps Jewish people remember the liberation of Israelites from Egypt. Connerton argues that such rituals reenact images of the past and that other types of physical behavior (e.g., dinner table etiquette, attire, and greeting gestures) also demonstrate and maintain collective memories. Thus, a future direction for psychological scientists interested in collective memory is to find ways to quantitatively measure and manipulate nonverbal memories.

A limitation of this research is that both subject populations (MTurk in E1 and college undergraduates in E2) are nonrepresentative convenience samples. MTurk samples are more diverse than college populations in key ways, yet they are also different from the population at large; for example, MTurk workers tend to be younger and more educated than the US population [27]. However, the correlation between overall recognition rates of the presidents across our two studies was high ($r$ = .76 [.60, .86]), suggesting a convergence between these two samples to explore in future work.

Some scientists also worry that MTurk workers may be more inattentive than other participant populations [28]. However, the majority of research indicates that MTurk samples generally replicate lab-based findings, even for rigorous multi-trial experiments, and even with some non-naïve participants [29, 30], especially when screening questions (e.g., did you use any external resources?) are used to exclude problematic subjects [31]. For this reason, and for the fact that only few studies have used MTurk to examine memory for US presidents, we find Experiment 1's MTurk data to be a reasonable starting point for better understanding how people in the US recognize the faces of historical leaders.

The current research shows that while both the faces and names of the US presidents are stored in American's collective memories, there is not a perfect one-to-one correspondence between these representations. It is unclear, though, whether subjects made their decisions via recollection of the presidents' faces or merely on their subjective perception of how "presidential" the faces appeared, suggesting another interesting line of future research.

At the time of this writing, Obama and Trump are some of the most well-known people in the world, by both face and name. However, it is almost inevitable that in 30 or 50 years, not only will they not be mentioned in the news anymore, but their faces and names will slowly fade from the public's collective memory, unless for whatever reason, they become central characters associated with distinctive and profoundly influential events in American history.

## Acknowledgments

We thank Alexa Rosenblatt, Jason Hayden, Will Deng, Sam Gary, and Hailey Rinella for their assistance with this project.

## Author Contributions

**Conceptualization:** Adam L. Putnam, Sarah Madison Drake, Serene Y. Wang, K. Andrew DeSoto.

**Data curation:** Adam L. Putnam.

**Formal analysis:** Adam L. Putnam, Serene Y. Wang.

**Methodology:** Adam L. Putnam, Sarah Madison Drake.

**Project administration:** Adam L. Putnam.

**Resources:** Adam L. Putnam.

**Software:** Adam L. Putnam, Sarah Madison Drake.

**Supervision:** Adam L. Putnam.

**Writing – original draft:** Sarah Madison Drake, Serene Y. Wang.

**Writing – review & editing:** Adam L. Putnam, Sarah Madison Drake, Serene Y. Wang, K. Andrew DeSoto.

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
