## [Decision Letter · Decision Letter 0]

20 Apr 2021

PONE-D-21-04866

Collective memory for American leaders:Measuring recognition for the names and faces of the US presidents

PLOS ONE

Dear Dr. Putnam,

Thank you for submitting your manuscript to PLOS ONE.  After careful consideration by two experts in the field and by myself, we feel that it has merit but could be upgraded in respect to two minor points that you are free to address in a revision of your work.  We are delighted to report that all reviewers have unanimously  recommended your research for publication.  However, we invite you to submit a revised version of the manuscript that addresses the following points raised during the review process:

(1) as requested by Reviewer 2 below, it would be helpful if you could provide some more discussion of your findings in the context of non-verbal aspects and mechanisms of collective memory, as this represents a much underinvestigated issue in the field and your particular novel contribution to that topic; 

(2) also, given your work on the pictorial aspects, could you please have a look and comment on any available data in your present research as to whether any priority/difference exists for the presidents shown on the US dollar banknotes or alike, because this could have eventually affected the results. 

Please submit your revised manuscript not later than six months from this date as beyond this time frame any revision has to be considered as a new submission. If you will need more time than this to complete your revisions, please reply to this message or contact the journal office at plosone@plos.org. Please include the following items when submitting your revised manuscript:

We look forward to receiving your revised manuscript.  Thank you for considering PLOS ONE for reporting your research. 

Kind regards,

Sasha

Alexander N. 'Sasha' Sokolov, Ph.D.

Academic Editor

PLOS ONE

Journal Requirements:

Reviewers' comments:

Reviewer's Responses to Questions

**Comments to the Author**

1. Is the manuscript technically sound, and do the data support the conclusions?

Reviewer #1: Yes

Reviewer #2: Yes

2. Has the statistical analysis been performed appropriately and rigorously? 

Reviewer #1: Yes

Reviewer #2: Yes

3. Have the authors made all data underlying the findings in their manuscript fully available?

Reviewer #1: Yes

Reviewer #2: Yes

4. Is the manuscript presented in an intelligible fashion and written in standard English?

Reviewer #1: Yes

Reviewer #2: Yes

5. Review Comments to the Author

Reviewer #1: This paper is a replication study of sorts that examines a well-documented finding about Americans' memory for presidents. Previous studies have demonstrated that there is a pattern to this form of collective memory that shows a serial position curve (highest memory for first few presidents and for most recent presidents with a spike in the middle due to the importance of Abraham Lincoln). These studies have been done with the names of presidents, but the current study does this with their pictures. At first glance, this replication may appear to add little to the research findings, but another robust finding in psychology is that memory for images is often impressively superior to that for words, suggesting that memory in this study might be superior to that found in others. This what gives the current study its interest and importance. To some degree this still does not amount to an earthshaking finding, but it does add an important piece to the overall puzzle of how collective memory functions, a topic that has been the topic of growing interest in psychology and other social sciences. The collection and presentation of the results are done in a very professional and organized manner, and the writing is very clear.

Reviewer #2: This is a well-designed extension of prior work on the serial position curve in memory for the US presidents from name recognition/recall to face recognition. The general findings of the prior literature are replicated, with primacy, recency, and Abe Lincoln effects, with the exception that overall, picture recognition is lower than name recognition. This finding would seem to be at odds with the picture superiority effect, but the authors make a plausible argument that the format in which people learn about the presidents is almost always verbal rather than pictorial. Thus, all else being equal, fluency of processing and familiarity will be higher for names than presidential portraits. This point is interesting, and might be expanded upon as the subject of future work (but is not necessary for the current manuscript). I don't believe there are any technical points on which I would like to see revision. If they were so inclined, the authors might spend a little more time discussing how collective memories may take different forms than the verbal, narrative form that has been focused on almost exclusively in the empirical research. This is of interest because non-verbal collective memories may behave by different rules than narrative collective memories (Paul Connerton, for instance, writes about collective memory as a sort of procedural memory, which is another departure from collective memory as declarative stories).

6. PLOS authors have the option to publish the peer review history of their article (what does this mean?). If published, this will include your full peer review and any attached files.

Reviewer #1: No

Reviewer #2: No

---

## [Author Response · Author response to Decision Letter 0]

28 May 2021

Dear Dr. Sokolov,

Thank you for the thoughtful comments from yourself and the two reviewers. We have revised the paper in light of the suggestions. Below we list your comments along with your suggested revisions, and our response to those suggestions.

Editor

Thank you for submitting your manuscript to PLOS ONE. After careful consideration by two experts in the field and by myself, we feel that it has merit but could be upgraded in respect to two minor points that you are free to address in a revision of your work. We are delighted to report that all reviewers have unanimously recommended your research for publication. However, we invite you to submit a revised version of the manuscript that addresses the following points raised during the review process:

(1) as requested by Reviewer 2 below, it would be helpful if you could provide some more discussion of your findings in the context of non-verbal aspects and mechanisms of collective memory, as this represents a much underinvestigated issue in the field and your particular novel contribution to that topic; 

Thank you for this valuable suggestion. We have added a paragraph to the general discussion that essentially makes the following point: while collective memory researchers from a variety of disciplines have conceptualized of collective memory as having many different forms (including verbal, imagery-based, and even procedural memory), currently quantitative approaches to studying collective memory in psychology have focused nearly exclusively on verbal memories. We argue that the current study provides a nice additional example of research examining non-verbal collective memories and recommend that future researchers continue to look to this extension.

(2) also, given your work on the pictorial aspects, could you please have a look and comment on any available data in your present research as to whether any priority/difference exists for the presidents shown on the US dollar banknotes or alike, because this could have eventually affected the results. 

This was another valuable suggestion. In the exploratory data analysis section for each experiment we report descriptive statistics examining whether an individual appearing on US currency affects presidential recognition or confidence in that decision. Briefly, both presidents and non-presidents that appear on US currency are more likely to be remembered as presidents, and those decisions are generally more confident, than individuals who do not appear on US currency. Due to the small number of observations in these categories (e.g., there are only two non-presidents who appear on currency) we refrained from conducting inferential statistics and instead just reported descriptive statistics.

In addition to the suggested revisions, we made a number of smaller grammar and style changes that we won’t bother to list here. Thank you in advance for considering this manuscript for publication in PLOS One.

---

## [Editor Report · Decision Letter 1]

11 Jun 2021

PONE-D-21-04866R1

Collective memory for American leaders:

Measuring recognition for the names and faces of the US presidents

PLOS ONE

Dear Dr. Putnam,

thank you for submitting your revised manuscript to PLOS ONE and addressing the minor comments of the Reviewers.  A PLOS ONE Staff Editor has drawn my attention to the fact that your research partly uses Amazon MTurk.  While PLOS ONE does considers such studies for publication and your work comprises other non-web-based samples, we would like to invite you to include a comment on potential limitations commonly associated with this type of study, namely, the non-naivety and trustworthiness of participants.  For more details, please see http://www.annualreviews.org/doi/abs/10.1146/annurev-clinpsy-021815-093623 and http://journals.plos.org/plosone/article?id=10.1371/journal.pone.0057410#s15.  I am not intending to send your updated manuscript out to the Reviewers. 

Please submit your revised manuscript within six months from this date as otherwise, any revision has to be considered a new submission.  If you will need more time than this to complete your revisions, please reply to this message or contact the journal office at plosone@plos.org. Please include the following items when submitting your revised manuscript:

A rebuttal letter that responds to each point raised by the academic editor and reviewer(s). You should upload this letter as a separate file labeled 'Response to Reviewers'.A marked-up copy of your manuscript that highlights changes made to the original version. You should upload this as a separate file labeled 'Revised Manuscript with Track Changes'.An unmarked version of your revised paper without tracked changes. You should upload this as a separate file labeled 'Manuscript

Thank you for choosing PLOS ONE for reporting your research.  We look forward to receiving your revised manuscript. 

Kind regards,

Sasha

Alexander N. 'Sasha' Sokolov, Ph.D.

Academic Editor

PLOS ONE
---

## [Author Response · Author response to Decision Letter 1]

25 Jun 2021

Thank you for the opportunity to address concerns related to online samples in the current set of studies. In response to your suggestion we added two paragraphs to the general discussion where we highlight potential concerns with using online samples (such as Amazon’s Mechanical Turk), and whether such concerns limit the generalizability of our results.

As suggested by the editorial office we also added a statement clarifying that our studies were approved by the Furman IRB

---

## [Editor Report · Decision Letter 2]

13 Jul 2021

Collective memory for American leaders: Measuring recognition for the names and faces of the US presidents

PONE-D-21-04866R2

Dear Dr. Putnam,

thank you for addressing the points raised in the previous email exchange.  I apologize for the delay in reply due to unexpected commitments, but I am happy to inform you that your manuscript has been deemed suitable for publication and will be formally accepted for publication once it meets all outstanding technical requirements.

Thank you for choosing PLOS ONE for communicating your research.

Best regards,

Sasha

Alexander N. 'Sasha' Sokolov, Ph.D.

Academic Editor

PLOS ONE
---

## [Editor Report · Acceptance letter]

16 Jul 2021

PONE-D-21-04866R2 

Collective memory for American leaders: Measuring recognition for the names and faces of the US presidents 

Dear Dr. Putnam:

I'm pleased to inform you that your manuscript has been deemed suitable for publication in PLOS ONE. Congratulations! Your manuscript is now with our production department. 

Kind regards, 

on behalf of

Dr. Alexander N. Sokolov 

Academic Editor

PLOS ONE